# Changes in the Microbial Community Diversity of Oil Exploitation

**DOI:** 10.3390/genes10080556

**Published:** 2019-07-24

**Authors:** Jingjing Liu, Jing Wu, Jiawei Lin, Jian Zhao, Tianyi Xu, Qichang Yang, Jing Zhao, Zhongming Zhao, Xiaofeng Song

**Affiliations:** 1Department of Biomedical Engineering, Nanjing University of Aeronautics and Astronautics, Nanjing 210016, China; 2School of Biomedical Engineering and Informatics, Nanjing Medical University, Nanjing 211166, China; 3Dalian Chivy Biotechnology Limited Company, Liaoning 116023, China; 4Center for Precision Health, School of Biomedical Informatics, The University of Texas Health Science Center at Houston, Houston, TX 77030, USA; 5Human Genetics Center, School of Public Health, The University of Texas Health Science Center at Houston, Houston, TX 77030, USA

**Keywords:** microbial community, 16S rRNA sequencing, microbial enhanced oil recovery technology

## Abstract

To systematically evaluate the ecological changes of an active offshore petroleum production system, the variation of microbial communities at several sites (virgin field, wellhead, storage tank) of an oil production facility in east China was investigated by sequencing the V3 to V4 regions of 16S ribosomal ribonucleic acid (rRNA) of microorganisms. In general, a decrease of microbial community richness and diversity in petroleum mining was observed, as measured by operational taxonomic unit (OTU) numbers, α (Chao1 and Shannon indices), and β (principal coordinate analysis) diversity. Microbial community structure was strongly affected by environmental factors at the phylum and genus levels. At the phylum level, virgin field and wellhead were dominated by *Proteobacteria*, while the storage tank had higher presence of *Firmicutes* (29.3–66.9%). Specifically, the wellhead displayed a lower presentence of *Proteobacteria* (48.6–53.4.0%) and a higher presence of *Firmicutes* (24.4–29.6%) than the virgin field. At the genus level, the predominant genera were *Ochrobactrum* and *Acinetobacter* in the virgin field, *Lactococcus* and *Pseudomonas* in the wellhead, and *Prauseria* and *Bacillus* in the storage tank. Our study revealed that the microbial community structure was strongly affected by the surrounding environmental factors, such as temperature, oxygen content, salinity, and pH, which could be altered because of the oil production. It was observed that the various microbiomes produced surfactants, transforming the biohazard and degrading hydro-carbon. Altering the microbiome growth condition by appropriate human intervention and taking advantage of natural microbial resources can further enhance oil recovery technology.

## 1. Introduction

Microorganisms play an important role in the biogeochemical cycle of the earth. The microbial community has revealed the interactions between earth and human activities, such as the forming and mining of petroleum fields [1]. Microbial communities could be influenced by factors such as temperature, oxygen content, salinity, and pH, which could be greatly altered during oil production. The interactions of these factors may cause changes in the overall abundance of bacteria and fungi. Subsequently, the community composition will be altered, leading to physical or chemical variation in the oil producing sites [2,3].

Microbial distribution and diversity in oil reservoirs have been widely studied during recent years. Microbiome is highly involved in microbial enhanced oil recovery (MEOR) technology [4]. The sulfate-reducing bacteria (SRB) can lead to microbiologically influenced corrosion (MIC) [5] and human intoxication, as it reduces sulphate to a hydrogen sulfide property. Accordingly, it is important to monitor and reduce SRB levels in an oil reservoir. Injection of nitrate and nitrate reducing bacteria (NRB) into the reservoir or activating endogenous NRB will inhibit the growth of SRB and achieve a biotransformation of hydrogen sulfide [6,7,8]. Crude emulsion formed in the process of petroleum production is one of the most important problems in the petroleum industry [9]. *Bacillus subtilis*, *Norcardia amarae*, *Rhodococcus aurantiacus,* and *Corynebacterium petrophilum* have been reported to produce biological surfactant that emulsifies heavy crude oils [10]. By manipulating the microbiome in an oil reservoir, many of the petroleum industry problems, such as MIC, human intoxication and oil emulsifying, can be solved. Furthermore, humans can take advantage of the knowledge of these microbes to improve oil production [11,12]. For example, *Pseudomonas*, *Achromobacter,* and *Thiobacillus* can convert heavy oil into light oil [13]. Partial bacteria called fermentative bacteria (FB) can use sugar, amino acids, long chain organic acids, and other organic compounds to produce natural gas, thereby increasing reservoir pressure, reducing crude oil viscosity, and increasing the fluidity of crude oil [14]. Thus the study of microbial distribution and diversity is important and beneficial to the oil industry.

Previous studies have mainly focused on the microbial community in crude oil or virgin field [15,16]. However, during the oil production, especially after water injection, the physical and chemical environment of an oil reservoir will affect the diversity of the microbial community. Kunio Yamane et, al. found that of 184 clones in total from the oil deposit libraries, 83 (45%) were represented by SRDP-BD05 and 71 (39%) were represented by SRDP-BA601 and were closely related to *Thermotoga hypogea*, while the phylotypes in the bacterial libraries from the wellhead were extremely diverse. Detected sequences in the wellhead were affiliated with *Gammaproteobacteria*, *Deltaproteobacteria*, *Epsilonproteobacteria*, *Clostridia*, *Bacilli*, *Bacteroidetes*, *Spirochaetes*, *Thermodesulfobacteria,* and *Thermotogae*. The most abundant bacterial group was *Thermotogae* in the oil deposit and *Epsilonproteobacteria* in the wellhead [17]. The population of some indigenous microorganisms might decrease or die under environmental changes. Such changes in populations of microbial inhabitants could have caused the difference between microbial communities in the wellheads and communities in deposits. To gain the knowledge of microbial community alteration before and after oil production, in this study we investigated the microbial communities in three sites: virgin field (VF), crude oil (oil-water mixture) collected from the wellhead (WH), and sludge collected from oil storage tanks (ST). Specifically, we studied the effect of petroleum exploitation on oil reservoir microorganism by performing 16S ribosomal nucleic acid (rRNA) sequencing. Our study aimed to undercover the microbial community diversity changes through oil production. Microorganisms that may benefit the oil industry have been discovered, and their symbiotic relationship has been revealed. Based on the correlation analysis result, *Bacillus* would inhibit SRB growth and cause moderate bacterial metal corrosion problems during oil storage. By analyzing the composition and distribution of the microbiome from different oil producing procedures, we hope the findings can help to develop improved approaches for increasing oil recovery and quality, as well as reducing oil pollution.

## 2. Materials and Methods

### 2.1. Sample Collection

Samples were collected from the virgin field, wellhead, and sludge of the same oil reservoir in Northeast China. The samples from the virgin field, wellhead, and storage tank sludge were labeled as VF, WH, and ST, respectively. Samples collected at the same site were used as replicates; for each sampling site, three replicates were prepared. The temperature of the oil reservoirs was between 46 °C and 52 °C. The salinity level of the oil reservoirs was at 1583–1934 mg/L. The genomic DNA of all samples were extracted and stored at −20 °C, as described in Li et al. [18].

All the 16S rRNA sequencing raw data can be downloaded in the National Genomics Data Center (BIGD) [19] by browsing accession ID CRA001794.

### 2.2. 16S rRNA Sequencing

The V3 to V4 region of the 16S rRNA gene (approximately 460 bp) was amplified with primer set 341F (5′-CCTACGGGNGGCWGCAG-3′) and 806R (5′-GGACTACHVGGGTWTCTAAT-3′). This primer pair could cover the V3–V4 region well [20]. Polymerase chain reaction (PCR) cycles were performed as follows: after 5 min of initial denaturation at 95 °C, followed by 30 cycles of 95 °C for 30 s, 55 °C for 30 s, 72 °C for 45 s, and an extension step at 72 °C for 10 min. The whole sequencing process was conducted by Shanghai Personal Biotechnology Co. (Shanghai, China) using an Illumina MiSeq platform (MiSeq 2 * 300 bp).

### 2.3. Data Analysis

Sequencing reads were screened using a sliding window (the size of the window was 10 bp, the step length was 1 bp), and read trimming was conducted. All reads shorter than 150 bp were discarded with no ambiguous base allowed. Forward and reverse reads with overlapping length larger than 10 were merged by FLASH (Fast Length Adjustment of SHort reads, College Park, MD, USA) (v1.2.7, http://ccb.jhu.edu/software/FLASH/) [21]. The resulting sequences were de-multiplexed and analyzed by using QIIME (v1.9.1) [22]. The primer sequence was further filtered with the maximum one mismatch and no ambiguous base calls. After discarding the consecutive base and removing chimeras by USEARCH, the clean sequence data were obtained [23].

Sequences were clustered into operational taxonomic units (OTUs) at 97% similarity using USEARCH [24]. The representative sequence was chosen based on the abundance and was aligned under a given taxonomic classification using the Greengenes database, and low abundance OTUs (lower than one sequence) were removed [25]. Rarefaction analysis was performed to assess the sequencing depth of each sample [26]. To compare the diversity of each sample, Chao1 and Shannon indices were calculated at the lowest sequencing depth by random sampling using QIIME [19]. Core and specific OTUs were calculated and illustrated by Venn diagrams (http://bioinformatics.psb.ugent.be/webtools/Venn/).

Principal coordinate analysis was used to visualize distance matrices and evaluate the global differences between samples. The top 50 genera and their abundance were illustrated in a heatmap generated by STAMP [27]. PICRUSt (Phylogenetic Investigation of Communities by Reconstruction of Unobserved States) was used to evaluate microbial community composition and metabolic transformation [28].

## 3. Results and Discussion

### 3.1. Microbial Community Abundance

Sample VF1 was discarded since 96.5% of the bacteria identified belonged to the *Acinetobacter* genus, which is a typical contamination phenomenon. After filtering, trimming and chimera checking, a total of 247,506 reads were detected. About 46.2% of the total OTUs were discarded during the removing singleton sequence process. Finally, 3439 OTUs and 244,558 reads were analyzed (Figure 1). The number of OTUs of each sample is shown in Table 1. Overall, VF group had the highest number of OTUs, while the ST group had the lowest number of OTUs. This difference indicated a microbe loss during oil production.

To assess species richness and compare microbiome diversity between different samples at the same level, rarefaction curves were plotted. From the curve, we observed that the OTU number tended to be stable as the reads number increased, which meant that the number of identified microbial species would not increase even if we used a larger sequence depth. Thus, the lowest sequence depth was chosen where the number of OTUs became a stable state (Figure 2). The Chao1 and Shannon indices, which reflect the species diversity, were generated by randomly sampling within the lowest sequencing range. VF3 had the highest species diversity (Chao1 index, 1352) while ST1 had the lowest Chao1 index with 201 (Figure 3A). The same tendency was also observed in the Shannon index, with the highest value in sample VF3 (7.51) and the lowest value in ST1 (1.17) (Figure 3B). The results above indicated that the VF group had the highest microbial community abundancy, while the ST group had the lowest, demonstrating microbial community loss through oil production.

### 3.2. Microbial Community Diversity Analysis at Phylum Level

Principal coordinate analysis was applied to reveal the heterogeneity of the microbial community between VF, WH, and ST (Figure 4). A tight clustering and complete separation revealed a totally different microbial community composition via sample groups. For each sample, the microbial community composition at the phylum level was shown in Figure 5. *Proteobacteria* was the most abundant phylum in the groups VF and WH (more than 50%), while it was the minority phylum in ST (2%). *Proteobacteria* abundance was lower in WH (48.6–53.4%) than in VF because of the lower temperature which was caused by water injection during oil production. *Proteobacteria* are anaerobic and chemolithoautotrophic; they survived better in VF and WH where they had less oxygen and sufficient organic carbon [29]. *Proteobacteria* was composed of *Alphaproteobacteria*, *Betaproteobacteria*, *Deltaproteobacteria*, *Epsilonproteobacteria,* and *Gammaproteobacteria* at class level (Appendix A). *Firmicutes* were observed in all sample groups and were the major phyla in ST, including *Bacilli*, *Clostridia,* and *Erysipelotrichia* (Appendix A) (larger than 60%). Storage tank sludge had less nutrition and a bad environment; *Firmicutes* could produce endospores and survive in extreme conditions [30]. However, *Firmicutes* became less competitive when the carbon source was sufficient, making *Proteobacteria* the dominant phylum in VF and WH. Based on our data, *Actinobacteria* was relatively abundant in ST and VF but was very limited in WH. On the contrary, *Acidobacteria* was abundant in WH, but limited in VF and barely presented in ST. The composition of *Actinobacteria* and *Acidobacteria* is shown in Appendix A, respectively. *Acidobacteria* was the third most abundant phylum in WH (10%). In contrast, it was the minority in VF (around 1%) and even absent in the ST group. This was because *Acidobacteria* are acidophilic, and the pH variation of VF, WH, and ST would affect *Acidobacteria* profiling [31]. The high abundance of *Acidobacteria* in WH suggested a more acidic environment.

There was a large number of mesophilic microorganisms present in WH and ST, which was also reported by Lin et al. [32]. Because of a long period of water injection, the oil reservoir temperature would decrease to below the original temperature, thus making an environment adaptable for mesophilic microorganisms. It is beneficial for MEOR that mesophilic microorganisms grow, since mesophilic microorganisms produce biosurfactant more easily than thermophilic bacteria.

### 3.3. Microbial Community Diversity Analysis at the Genus Level

To uncover the diversity of the microbial community, the community composition was analyzed at the genus level. Fifty genera with the highest sequence read number were selected for further examination of the microorganism community composition. The fifty genera and their related abundance were illustrated in a heatmap plotted by STAMP (Figure 6). The dendrogram on top suggested a global variance of community composition between sample groups [33]. The predominant genera in VF were *Ochrobactrum* and *Acinetobacter*. *Ochrobactrum* and *Acinetobacter* were capable of producing biosurfactant, which is beneficial for recovering oil from an oil-saturated core. Furthermore, *Acinetobacter* was efficient in degrading heavy components to light components, and it has been described as a halophilic oil-utilizing and rhamnolipid-producing bacteria [34,35]. In WH, the major microbial communities were *Lactococcus* and *Pseudomonas*. *Lactococcus* produces biosurfactants and has been used in the oil recovery, drilling, bioremediation, and the removal of heavy metal contaminants [36]. *Pseudomonas* was an important petroleum hydrocarbon degrader and was applied in oil pollution remediation [37]. *Prauseria* and *Bacillus* were abundant in ST. Both of them have the potential of producing active surfactant compounds and bio-polymer and also permeability modification. In addition, *Bacillus* is useful for green energy production because it can remove nickel and sodium from crude oil. Furthermore, *Bacillus* can reduce nitrite or nitrate compounds and thus inhibit the growth of SRB and achieve the biotransformation of hydrogen sulfide which is poisonous to humans [38,39].

The correlations between the top 50 genera were investigated (Figure 7). The Spearman rank correlation coefficient of the 50 genera was calculated using Mothur. The correlation coefficient and *p*-value were set at 0.8 and 0.01, respectively, for the correlation network construction. The correlation networks of these genera were roughly divided into two clusters—the genera in the left cluster were positively correlated and *Sphingomonas* was the only genus that related to the other genera in the right cluster. *Pseudonocardia* and *Bacillus* were negatively correlated to the other genera, which suggests a certain competition between the two and other genera.

Statistical analysis (analysis of variance) was conducted to locate the significant differentially abundant genera (*p*-value <0.01, fold change >4), and the genera distribution within sample groups are illustrated in Figure 8. VF and WH had more than 10 unique genera each. The unique genera in VF were thermophilic, anaerobic bacteria. Most of them were proven to be originating from the animal or human gut or feces [40]. The unique genera in WH were from the natural environment, with several genera having nitrogen fixation properties [41]. Genera that were unique or changed significantly at different sampling sites are listed in Table 2. *DA101*, *Carnobacterium*, *Burkholderia*, and *Prauseria* were the four genera dramatically changed via the sample group. Compared with VF, the abundance of *DA101* decreased dramatically in WH. The possible reason is that *DA101* prefers elevated amounts of labile carbon input, indicating a higher organic carbon content in VF than WH [42]. The relatively high abundance of *Carnobacterium* in WH may have been caused by water injection during oil production, since water from outside brings in a large amount of *Carnobacterium* [43]. The anaerobic and organic nitrogen degrading properties of *Burkholderia* made it rich in VF and WH, while limited in ST [44]. The abundance of *Prauseria* tremendously increased in ST, indicating a lower salinity content in ST than VF and WH, since *Prauseria* has been reported to prefer a low salt environment [45].

In this study, microbial communities in different stages of oil production were studied using sequencing. The results revealed that the reservoir harbored diverse microbial populations. Based on 16S rRNA sequencing, a total of 58 bacterial phyla and 830 genera were found in the reservoir. The virgin field had the highest community diversity and species abundance compared with the other two sites that were investigated. Oil reservoir exploitation greatly changed the geological and chemical properties of the ambient environment, leading to a reduction of both diversity and abundance in the microbial community. Also, because of the alternation of externalities, the microbial composition changed greatly. In summary, our study found that the oil reservoirs contained rich species of indigenous microbes, and some species had active surfactant production, biohazard transformation, and hydro-carbon degradation functions. During oil exploitation, booming microbes that could produce surfactants would increase oil recovery, such as *Lactococcus*, *Pseudomonas*, *Prauseria, and Bacillus*. *Acinetobacter* was able to degrade heavy components to light components and *Lactococcus* could remove heavy metal; taking advantage of these microbes will benefit the oil quality. *Pseudomonas* was an important petroleum hydrocarbon degrader and could be applied in oil pollution remediation. SRB could cause metal corrosion and should be inhibited during oil storage. *Bacillus* is negatively correlated to SRB, thus by introducing *Bacillus*, this would inhibit SRB growth. Taking advantage of these microbiomes may help develop better strategies for using natural microbial resources in MEOR technology.

### 3.4. Assessment of Microbial Community Function

The microbial community was greatly diverse among the different oil producing sites. We applied pathway enrichment analysis of the gene sequences to identify enriched metabolic pathways, aiming to better understand the activities of the microbiome (Figure 9). Put briefly, carbohydrate metabolism and amino acid metabolism were the most enhanced metabolic pathways. These two enriched pathways have a rationale, since the oil reservoir was short of nutrition, and the microbiome enhanced the energy and fundamental material-producing pathways to better survive in the harsh environment. Metabolic pathways that were not crucial for survival were maintained at a relatively low level. For example, biosynthesis of the secondary metabolites was the most inconspicuous one among all the metabolic pathways. In most cases, ST had a slightly higher metabolic level than VF and WH, which was probably correlated to nutrition supply. No significant difference of metabolic pathways was observed between sample groups. It is worth noting that WH had a higher glycan biosynthesis and metabolism level than VF and ST. Glycan is crucial in proper protein folding and cell–cell interactions. Higher glycan biosynthesis and metabolism in WH reveals the instability and mutability of microbial communities [46]. Furthermore, according to cellular processes analysis, higher cell motility was also observed in WH. Superior motility properties means superior growth kinetic properties, suggesting a future growth boom in WH [47]. This result is in accordance with high glycan biosynthesis and metabolism in WH. Taken together, our results implied a future bloom in the community diversity of WH.

## 4. Conclusions

The environmental variation (temperature, oxygen content, pH value, and salinity) affects the diversity of microbial composition and community structures during oil reservoir production. Some species had active surfactant production, biohazard transformation, and hydro-carbon degradation functions which are beneficial to the oil industry. The deployment of MEOR technology should be determined according to the environmental characteristics and microbial communities to improve oil recovery and quality. The well-studied microbial community of different oil producing sites may help us understand microbiome and better serve MEOR technology.

## Figures and Tables

**Figure 1 genes-10-00556-f001:**
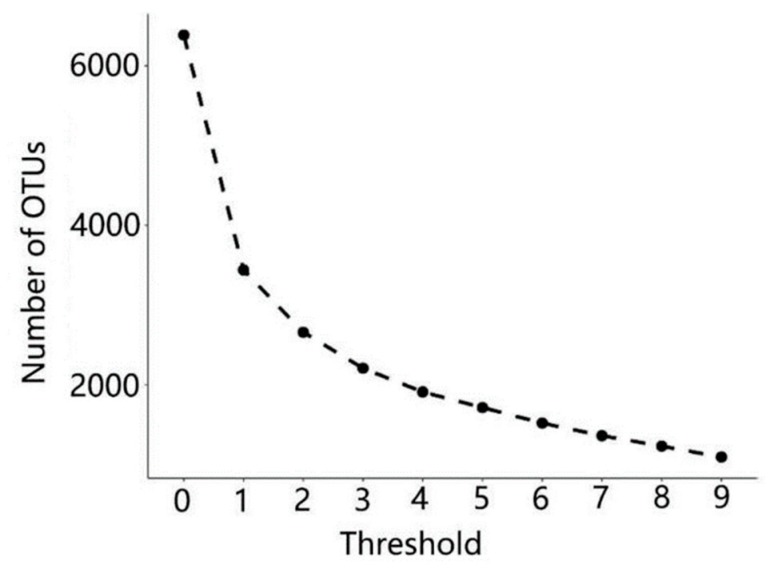
The number of operational taxonomic units (OTUs) under different filtering strategies. Threshold is the minimum number of sequences belonging to a single OTU.

**Figure 2 genes-10-00556-f002:**
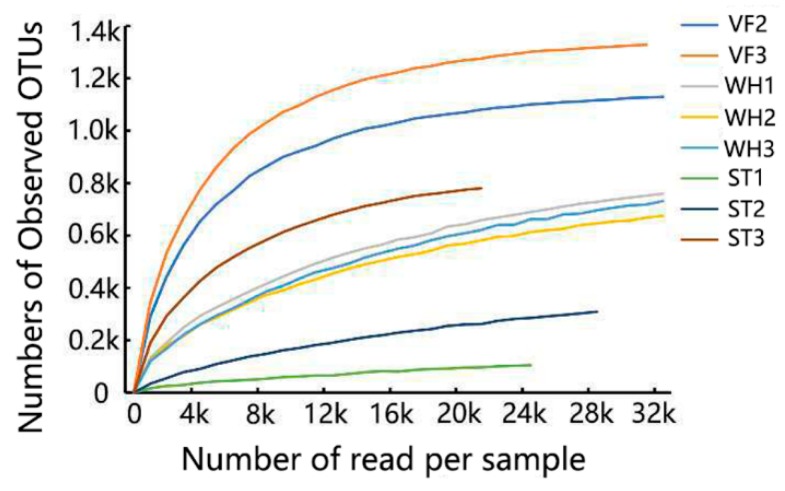
Rarefaction curve of VF, WH, and ST.

**Figure 3 genes-10-00556-f003:**
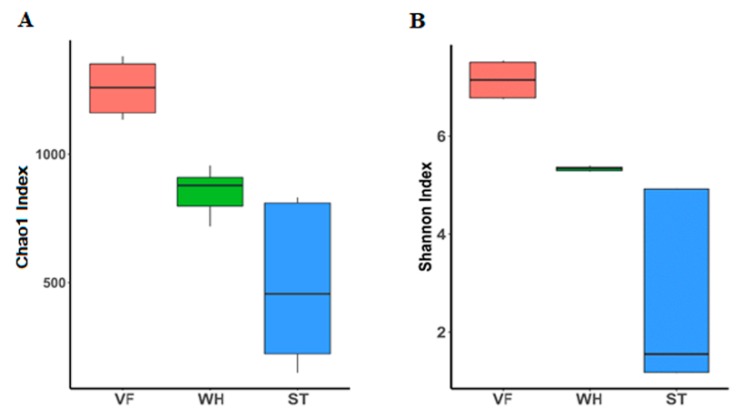
α diversity analysis of microbial communities in VF, WH, and ST. (**A**) Boxplot of the Chao1 index, which was calculated based on the abundance of each OTU (operational taxonomic unit, 97% cutoff). (**B**) Boxplot of Shannon index, which was calculated based on the abundance of each OTU.

**Figure 4 genes-10-00556-f004:**
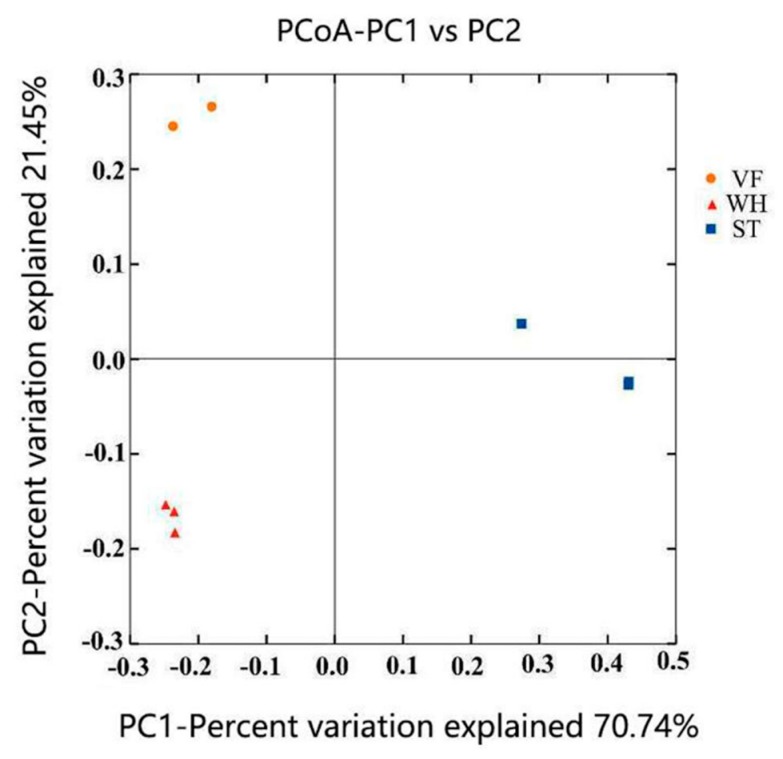
Principal coordinate analysis (PCoA) of the microbial community in VF, WH, and ST, based on the weighted UniFrac distance matrix.

**Figure 5 genes-10-00556-f005:**
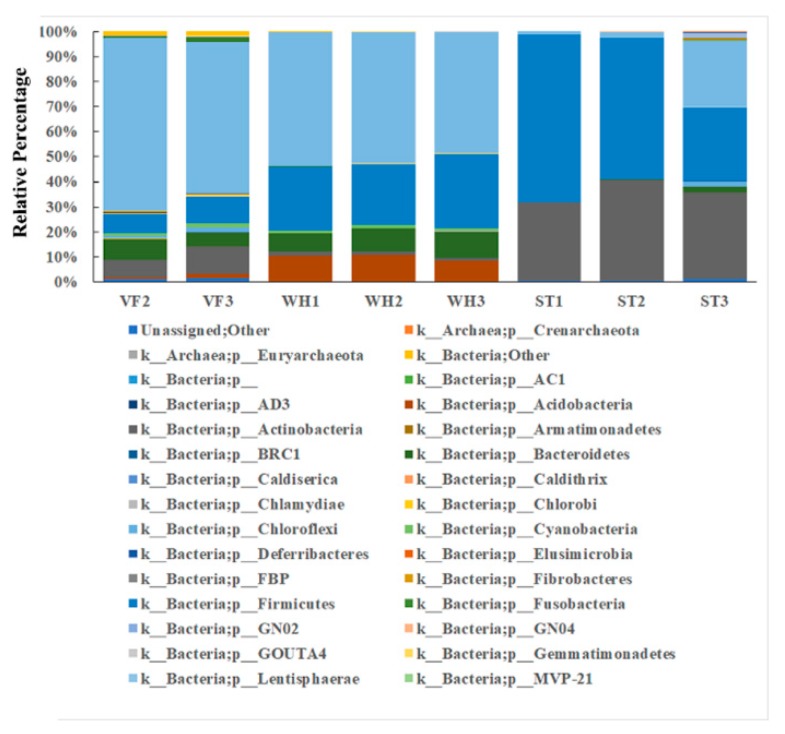
Relative abundance of the microbial communities in VF, WH, and ST, revealed by the 16S rRNA gene at the phylum level. The relative abundance is defined as a percentage of the total microbial sequences in a sample.

**Figure 6 genes-10-00556-f006:**
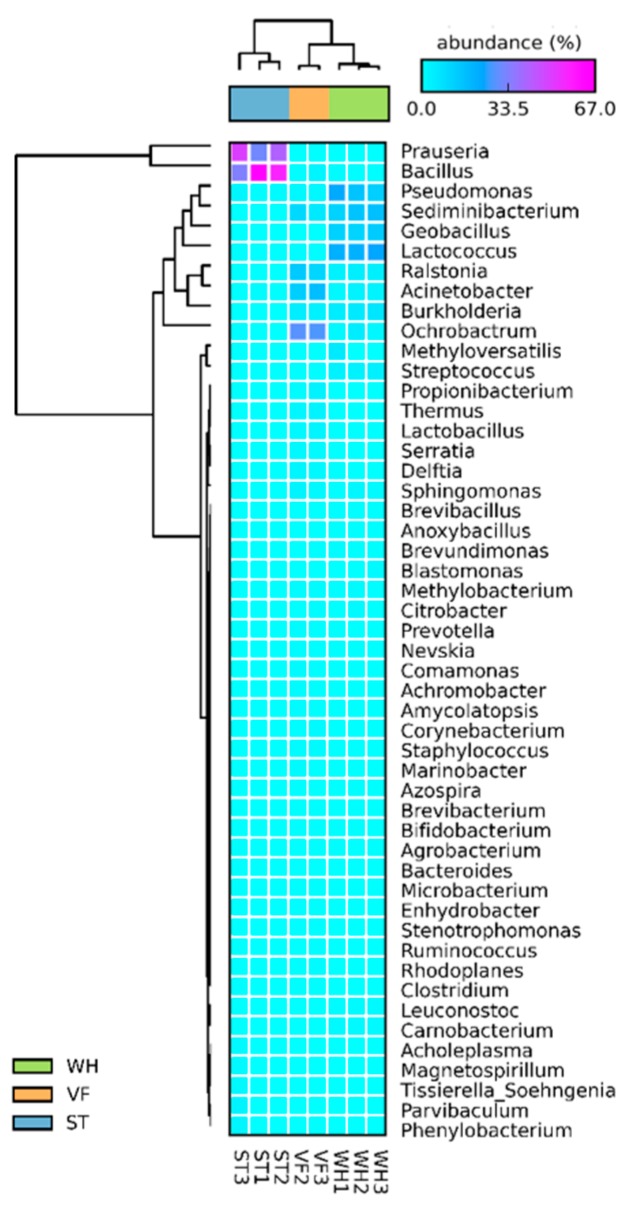
Heat map of the 50 most abundant genera in VF, WH, and ST. For each sample, the number of reads per OTU was normalized by total number of reads from the sample.

**Figure 7 genes-10-00556-f007:**
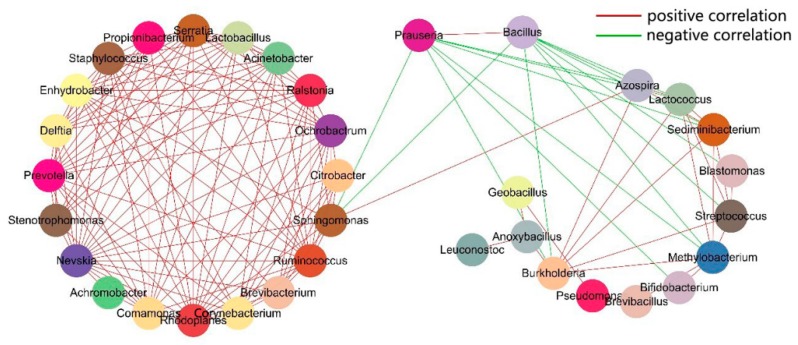
Correlation analysis of the 50 most abundant genera in VF, WH, and ST.

**Figure 8 genes-10-00556-f008:**
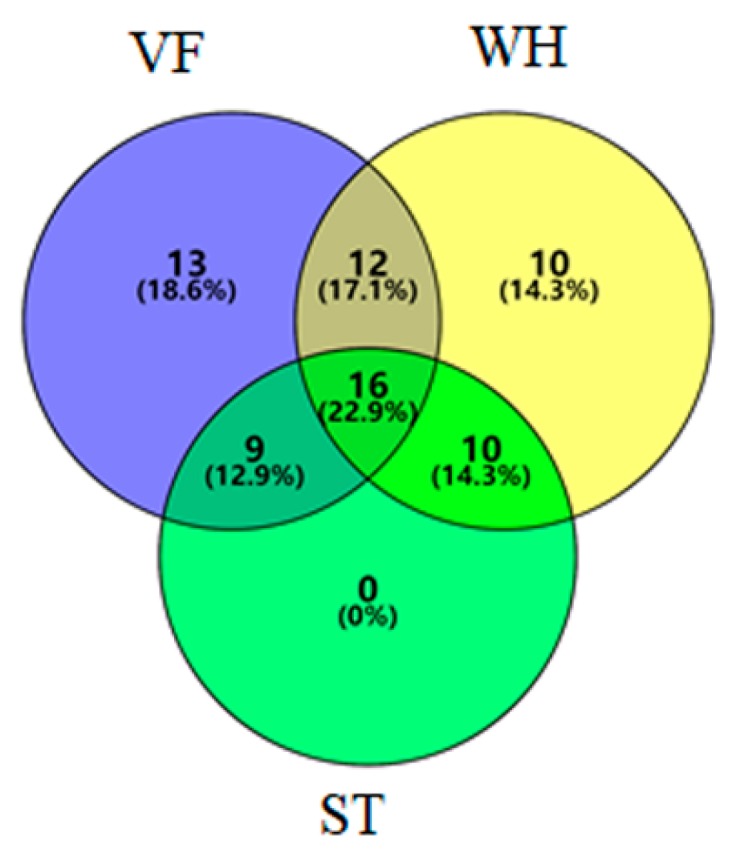
The distribution of the differentially abundant genera at the genus level in VF, WH, and ST.

**Figure 9 genes-10-00556-f009:**
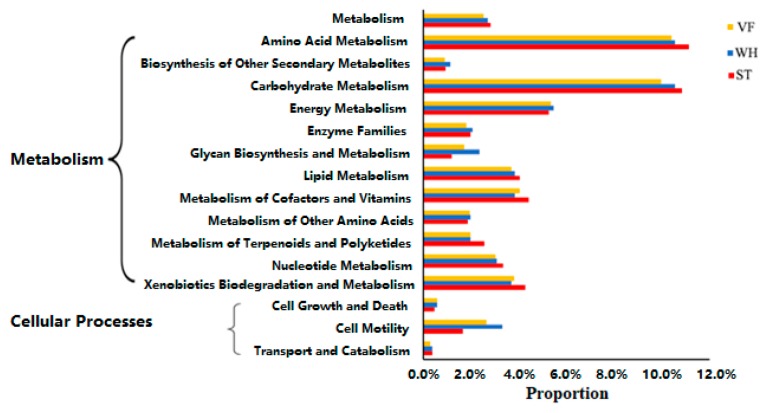
Phylogenetic investigation of communities by Reconstruction of Unobserved States (PICRUSt), predicting functional profile alteration of microbial communities enriched in VF, WH, and ST.

**Table 1 genes-10-00556-t001:** Statistics of MiSeq sequencing of 16S rRNA. VF: virgin field, WH: wellhead, ST: storage tank.

Sample ID	Number of Reads	Observed OTUs
VF2	33,817	1133
VF3	31,517	1328
WH1	33,007	764
WH2	33,743	684
WH3	39,734	774
ST1	24,608	107
ST2	28,311	309
ST3	22,769	784

**Table 2 genes-10-00556-t002:** Unique and differentially abundant genera within VF, WH, and ST.

**Unique genera**	**Unique Genera in VF**				**Unique Genera in WH**			
	***Caldilinea***				***Brevibacillus***			
	***Coprobacillus***				***Brochothrix***			
	***Nitrosovibrio***				***Butyrivibrio***			
	***Parachlamydia***				***Capnocytophaga***			
	***Sutterella***				***Bradyrhizobium***			
	***Syntrophobacter***				***Candidatus Rhabdochlamydia***			
	***Veillonella***				***Chryseobacterium***			
	***Acidovorax***				***GOUTA19***			
	***Allobaculum***				***Hydrogenophilus***			
	***Faecalibacterium***				***Nevskia***			
	***Leptotrichia***							
	***Turicibacter***							
	***YRC22***							
**Differentially abundant genera**	**Name**	**Type**	**Fold Change**	***p*-value**	**Name**	**Type ^1^**	**Fold Change**	***p*-value**
	*DA101*	VF/WH	87	5.22 × 10^−5^	*Lactobacillus*	VF/ST	81.5	4.84 × 10^−3^
	*Thermus*	VF/WH	68.2	4.89 × 10^−4^	*Sphingobacterium*	VF/ST	81	7.25 × 10^−5^
	*Anaeromyxobacter*	VF/WH	35.3	1.88 × 10^−3^	*Oscillospira*	VF/ST	59.6	8.38 × 10^−4^
	*Methylibium*	VF/WH	34.9	9.80 × 10^−4^	*Stenotrophomonas*	VF/ST	53.4	1.19 × 10^−4^
	*Adlercreutzia*	VF/WH	20.4	1.58 × 10^−3^	*Devosia*	VF/ST	45.8	9.82 × 10^−3^
	*Exiguobacterium*	VF/WH	16.5	2.59 × 10^−3^	*Streptomyces*	VF/ST	34.5	1.60 × 10^−3^
	*Deinococcus*	VF/WH	15.7	9.48 × 10^−3^	*Akkermansia*	VF/ST	31.5	1.00 × 10^−2^
	*Nitrospira*	VF/WH	14.8	1.56 × 10^−4^	*Methylobacterium*	VF/ST	24.5	1.87 × 10^−3^
	*Oscillospira*	VF/WH	14	8.31 × 10^−4^	*Planomicrobium*	VF/ST	21	3.15 × 10^−3^
	*Psychrobacter*	VF/WH	13.5	1.25 × 10^−3^	*Acinetobacter*	VF/ST	19.3	4.10 × 10^−4^
	*Desulfovibrio*	VF/WH	10.5	8.54 × 10^−3^	*Brevibacterium*	VF/ST	17.3	6.32 × 10^−3^
	*Clostridium*	VF/WH	7.5	8.28 × 10^−4^	*Prauseria*	VF/ST	1.15 × 10^−4^	2.13 × 10^−3^
	*Ochrobactrum*	VF/WH	7.1	3.11 × 10^−3^	*Burkholderia*	WH/ST	3708	6.76 × 10^−4^
	*Rhodanobacter*	VF/WH	5.4	1.28 × 10^−3^	*Sediminibacterium*	WH/ST	650.1	9.29 × 10^−4^
	*Staphylococcus*	VF/WH	4.5	5.12 × 10^−3^	*Ralstonia*	WH/ST	154.6	3.44 × 10^−4^
	*Stenotrophomonas*	VF/WH	4.5	8.16 × 10^−3^	*Carnobacterium*	WH/ST	123.9	5.18 × 10^−5^
	*Acinetobacter*	VF/WH	4.1	1.07 × 10^−3^	*Geobacillus*	WH/ST	109.2	2.11 × 10^−4^
	*Bacteroides*	VF/WH	3.9	3.22 × 10^−3^	*Lactococcus*	WH/ST	89.3	8.00 × 10^−5^
	*Microbacterium*	VF/WH	3.3	5.81 × 10^−3^	*Fusobacterium*	WH/ST	82	5.36 × 10^−3^
	*Anoxybacillus*	VF/WH	0.2	3.45 × 10^−4^	*Streptococcus*	WH/ST	81.4	1.98 × 10^−4^
	*Streptococcus*	VF/WH	0.1	2.79 × 10^−3^	*Leuconostoc*	WH/ST	65.8	2.59 × 10^−3^
	*Lactococcus*	VF/WH	3.09 × 10^−2^	1.09 × 10^−3^	*Staphylococcus*	WH/ST	51	1.33 × 10^−3^
	*Geobacillus*	VF/WH	1.12 × 10^−2^	2.13 × 10^−3^	*Methylobacterium*	WH/ST	35	4.15 × 10^−3^
	*Carnobacterium*	VF/WH	1.73 × 10^−3^	7.21 × 10^−4^	*Janthinobacterium*	WH/ST	32.5	1.59 × 10^−4^
	*Burkholderia*	VF/ST	2068.5	7.80 × 10^−3^	*Ruminococcus*	WH/ST	27.5	1.43 × 10^−4^
	*Thermus*	VF/ST	386.5	4.20 × 10^−4^	*Arthrobacter*	WH/ST	23.5	4.92 × 10^−4^
	*Ralstonia*	VF/ST	372.7	7.16 × 10^−3^	*Trueperella*	WH/ST	22	7.14 × 10^−3^
	*Deinococcus*	VF/ST	235.5	7.15 × 10^−3^	*Lactobacillus*	WH/ST	20.5	1.83 × 10^−3^
	*Staphylococcus*	VF/ST	231	2.35 × 10^−3^	*Ruminococcus*	WH/ST	18	1.00 × 10^−2^
	*Rhodococcus*	VF/ST	141	7.17 × 10^−3^	*Lysobacter*	WH/ST	17	7.60 × 10^−4^
	*Enhydrobacter*	VF/ST	114.8	7.95 × 10^−3^	*Pseudomonas*	WH/ST	15	4.74 × 10^−3^
	*Ochrobactrum*	VF/ST	100	1.30 × 10^−3^	*Anoxybacillus*	WH/ST	10.1	1.41 × 10^−3^

^1^ Represents way of obtaining ratio between sample groups.

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
