# Peer review of "Changes in the Microbial Community Diversity of Oil Exploitation"

_genes, 2019, doi:10.3390/genes10080556_

Round 1
Reviewer 1 Report
This is a decent text, ready for publication. It needs only minor check throughout the manuscript of the correctness of the names of bacteria and abberrations as well as taxonomic terminology (e.g. there are errors in the text like incorrect form „genuses”, errors in some bacteria names).
Author Response
Thanks for the comments and advices. We replaced “genuses” with “genera”, and corrected the bacteria names, for example, Proteobacteria in line 26, Erysipelotrichia in line 171.

Reviewer 2 Report
Please find comments attached in a separate document.

Author Response
Thanks for your comments and advice. Please find our responses to your questions in the attached file.

Round 2
Reviewer 2 Report
Thank you for addressing all questions and request appropriately. I agree to accept the manuscript in its present form, and have no further comments except for the potential further improvement of the English language.